# Profiling the health-related physical fitness of Irish adolescents: A school-level sociodemographic divide

**Brendan T. O'Keeffe**[1,2☉]\*, **Ciaran MacDonncha**[1,2☉], **Helen Purtill**[2,3☉], **Alan E. Donnelly**[1,2☉]

**1** Department of Physical Education and Sport Sciences, University of Limerick, Limerick, Ireland, **2** Health Research Institute, University of Limerick, Limerick, Ireland, **3** Department of Mathematics and Statistics, Faculty of Science and Engineering, University of Limerick, Limerick, Ireland

☉ These authors contributed equally to this work.
\* Brendan.okeeffe@ul.ie

## Abstract

### Background and aims

Examining factors that may explain disparities in fitness levels among youth is a critical step in youth fitness promotion. The purpose of this study was twofold; 1) to examine the influence of school-level characteristics on fitness test performance; 2) to compare Irish adolescents' physical fitness to European norms.

### Methods

Adolescents (n = 1215, girls = 609) aged 13.4 years (SD .41) from a randomised sample of 20 secondary schools, stratified for gender, location and educational (dis)advantage, completed a series of field-based tests to measure the components of health-related physical fitness. Tests included: body mass index; 20 metre shuttle run test (20 m SRT); handgrip strength; standing broad jump (SBJ); 4 x 10 metre shuttle run; and back-saver sit-and-reach (BSR).

### Results

Overall, boys outperformed girls in all tests, aside from the BSR ($p < 0.005$, t-test, Bonferroni correction). Participants in designated disadvantaged schools had significantly higher body mass index levels ($p < 0.001$), and significantly lower cardiorespiratory endurance (20 m SRT) ($p < 0.001$) and muscular strength (handgrip strength) ($p = 0.018$) levels compared to participants in non-disadvantaged schools. When compared to European norms, girls in this study scored significantly higher in the 20 m SRT, 4 x 10 metre shuttle run and SBJ tests, while boys scored significantly higher in the BSR test (Cohen's d 0.2 to 0.6, $p < 0.001$). However, European adolescents had significantly higher handgrip strength scores (Cohen's d 0.6 to 0.8, $p < 0.001$).

**Data Availability Statement:** All relevant data are within the manuscript and its Supporting Information files.

**Funding:** Initials: BOK Grant number: GOIPG/2017/789 Grant fund: Government of Ireland Postgraduate Scholarship Grant fund website: http://research.ie/ No role played by funders in any part of the study.

**Competing interests:** The authors have declared that no competing interests exist.

## Conclusion

Irish adolescents compared favourably to European normative values across most components of HRPF, with the exception of muscular strength. School socioeconomic status was a strong determinant of performance among Irish adolescents. The contrasting findings for different fitness components reiterate the need for multi-component testing batteries for monitoring fitness in youth.

## Introduction

Physical fitness is a multifaceted construct that can be described as an integrated measure of most, if not all, body functions that are involved in daily physical activity [1]. Health-related physical fitness (HRPF) is made up of multiple components including, cardiorespiratory endurance (CRE), musculoskeletal fitness (muscular strength, endurance, and power) and body composition, which have been identified as powerful markers of future health among children and adolescents [1, 2]. There is a consistent body of evidence supporting the favourable effects of moderate-to-high levels of physical fitness to health-related outcomes, including cardio-metabolic risk factors [3], musculoskeletal [4] and cognitive [5] traits in childhood and adolescence. It has also been reported that positive changes to HRPF during childhood and adolescence can mitigate the impact of negative health outcomes later in life [6].

Declining HRPF levels among youth internationally have been reported. For example, an international analysis of secular trends of CRE among adolescents, involving 11 countries from 1980 to 2000, noted a sample-weighted mean decline of 0.43% per year, and the decline was most prevalent in older adolescent age groups [7]. A meta-analysis of 20 m shuttle run test scores among a sample of 1,142,026 children and youth from 50 countries reported that 67% of boys (CI 95% ± 14%) and 54% (CI 95% ± 17%) of girls had healthy CRE according to Fitnessgram criterion referenced standards, and the numbers achieving healthy standards decreased systematically with age [8]. In contrast, Moliner-Urdiales and colleagues [9] reported significant increases in CRE among Spanish adolescents between 2001 and 2007 (Cohens d 0.2 to 0.4, $p < 0.05$). In terms of muscular fitness, Cohen et al. [10] reported that English schoolchildren have shown a decrease in upper body muscular strength, measured by handgrip dynamometer over the past decade, a trend also reported in Spain [9], Canada [11] and China [12]. However, Huotari et al. [13] reported that muscular fitness was higher in a cross sectional cohort of Danish adolescents in 2001 than an age-matched cohort from 1976.

Examining variations in HRPF data gathered from multiple countries should be interpreted with caution. Firstly, contrasting methodological approaches used for the measurement and interpretation of results can produce different outcomes [14]. Welsman and Armstrong argue that, in contrast to secular declines in cardio-respiratory fitness estimated from field-based measures such as those presented by Tomkinson et al. [7], lab-based measurement techniques reveal little or no differences in aerobic fitness among youth across time. Furthermore, minor changes to administration protocols, such as the measurement of handspan for handgrip strength as recommended by Ruiz and colleagues [15], could also result in measurement error when comparing scores across regions. The risks associated with criterion values have also been highlighted by Mahar et al. [16] who reported that 35% of 4th- and 5th-grade girls who achieved PACER standards failed to pass the 1-mile run/walk standards. Zhu et al. [17] have proposed the use of test equating statistics to enable two or more tests that measure the same construct in different ways to be compared on the same scale. Encouragingly, a consensus on

reliable and valid approaches to monitoring HRPF in field-based settings has emerged in recent years, as reflected in the recent National Academy of Medicine's report on measurement and health outcomes in youth [18], and the pan-European ALPHA (Assessing Levels of Physical Activity) project [19] that established a standardised test battery for monitoring HRPF throughout Europe.

At an individual level, variations in physical fitness are caused by a network of social, behavioural, physical, psychosocial and physiological factors [20]. Well established determinants of physical fitness among youth include age, gender and physical activity levels [21]. Inverse associations between physical fitness, particularly CRE, and overweight in adolescents have also been reported [21, 22]. The relationship between socioeconomic status and physical fitness has been less examined, with much of the research to date producing inconsistent results [23]. In addition, despite the prominence of fitness testing in schools [24], the influence of school-level sociodemographic characteristics including location and educational (dis)advantage on HRPF are scantly represented in the current literature. In one of the few studies to examine school sociodemographic characteristics and health nationally, Bel-Serrat and colleagues [25] reported that school-level educational (dis)advantage was a strong determinant of overweight and obesity in schoolchildren. Bai et al. [26] also concluded that there was clear evidence showing that school socioeconomic status was the most influential contextual factor for explaining disparities in school fitness outcomes among 157,971 schoolchildren from 675 schools in the US.

It has been projected that the Republic of Ireland is on course to become the most obese nation in Europe by the year 2030 [27]. Despite World Health Organisation recommendations [28], the Republic of Ireland lacks a clearly specified strategy for monitoring HRPF in youth. Consequently, there is a paucity of data on objectively measured fitness levels of Irish youth, with much of the health and activity surveillance surveys to date utilising self-reported measures [29]. To the authors' knowledge, the Children Sport Participation and Physical Activity study [30] is the only study to measure HRPF among a nationally representative sample of adolescents in the Republic of Ireland. Woods and colleagues [30] reported no significant changes in CRE levels between 2010 and 2018, with 76% and 77% of participants, respectively, meeting established criterion referenced standards [31]. The collection of objective measures of health and physical fitness from population-based samples over pre-defined time periods is a crucial resource that can inform policy-makers and the public, and is vital for healthcare and education authorities for timely planning of prevention programs [32]. In light of the scarcity of research specific to the Irish context, the aim of the current study was twofold. Firstly, to examine the influence of school-level characteristics on fitness test performance of Irish adolescents from a randomised and stratified sample of schools, and secondly, to compare these data to established European normative values [33].

## Methods

### Sampling and recruitment

Research ethics approval for this study and the associated protocols was granted by the Institution Review Board of the Faculty of ****, ***, *** (***). All secondary schools with access to an indoor hall space of $\geq$ 25 metres and students in year one of secondary school with no inhibiting health conditions were eligible to participate. A randomised sample of 20 schools, stratified for gender (boys' schools, girls' schools and mixed-gender schools), location (urban and rural categorised by population density), and educational (dis)advantage, participated in the study. Designated disadvantaged schools were selected based on Department of Education and Skills categorisations as part of the Government of Ireland's Delivering Equality of opportunity in

Schools (DEIS) scheme [34]. This classification is based on a 'Deprivation Index Scale' which accounts for demographic growth, social class composition and employment status, in addition to centrally held Department of Education and Skills pupil data. There are currently 185 designated disadvantaged secondary schools in Ireland, representing just over one quarter of all secondary schools [34]. The procedure for generating a randomised sample was conducted using a special computerized code system in which all secondary schools in the mid-west and south-west region of Ireland were assigned a code and categorised according to the aforementioned strata. Due to the geographical spread of schools, and the need to visit each school individually, 20 schools was considered to be the maximum sample size achievable from a logistical viewpoint, and the minimum required to obtain a sufficient number of schools in each of the chosen strata. Although, the randomised and stratified sample represented the largest review of HRPF among Irish adolescents undertaken to date, participants were generated from schools in the mid-west and south-west regions of Ireland only due to logistical constraints, and thus, findings cannot be generalised for the entire country.

**School and participant recruitment.** Of the initial sample of 20 schools, two schools were deemed ineligible to participate due to insufficient indoor hall space, and two schools declined to participate due to time constraints. In each case, the next school on a randomised reserve list was recruited. Approval from the principal and cooperating physical education teacher in each school was granted following an initial email and telephone conversation. Study information sheets and consent forms were provided by cooperating teachers to all students and their parents. Cooperating physical education teachers were responsible for gathering consent forms. This study focused specifically on students in year one of secondary school education (ages 13 to 14), and was open to all students in the selected year group in each participating school who provided informed consent to participate and fulfilled the physical activity readiness questionnaire (PAR-Q) [35] pre-test requirements. A total of 27 students were deemed ineligible to participate due to underlying health conditions recorded on the PAR-Q. A minimum participation rate threshold of 70%, as used in other similar studies [36], was set for a school to be considered eligible. Reasons for non-participation were recorded on a non-participant form. The most commonly cited reasons were absenteeism, injury/sickness, and/or the students or parents deciding not to provide consent to participate in the study. Participation rates in the final sample were $\geq$ 75% in each school, with a mean participation rate of 61 students per school. Testing took place over a three month period between November 2018 and January 2019. A demographic profile of participants is provided in Table 1.

**Table 1. Demographic profile of participants.**

| Category | Sub category | Number of schools | Participants (%) |
|---|---|---|---|
| Gender (participants) | Girls (Age: 13.4, SD .40) | - | 609 (50.1%) |
| | Boys (Age: 13.5, SD .43) | - | 606 (49.9%) |
| School gender | Boys | 4 | 47 (14.4%) |
| | Girls | 4 | 67 (20.5%) |
| | Mixed-gender | 12 | 213 (65.1%) |
| Educational (dis)advantage [a] | Non-disadvantaged | 14 | 994 (81.8%) |
| | Designated disadvantaged | 6 | 221 (18.2%) |
| Location [b] | Rural | 8 | 390 (32.1%) |
| | Urban | 12 | 825 (67.9%) |

[a] This classification is based on a 'Deprivation Index Scale' which accounts for demographic growth, social class composition and employment status, in addition to centrally held Department of Education and Skills pupil data.

[b] Categorised by population density: Urban, the cities of Cork and Limerick; Rural, all other areas of the mid and south west of the Republic of Ireland

## Testing procedures

The cooperating physical education teacher in each school selected eight senior students (final two years of secondary education) to facilitate the administration of the test battery. Tests were delivered in a station format to small groups of five students or less, and each administrator was responsible for one test item on the test battery. A manual detailing standard operating procedures for each test item was designed for and read by both cooperating teachers and senior student administrators. Cooperating physical education teachers and student facilitators participated in a three hour workshop in which each administrator was assigned one test, and trained in the assigned test only. A comprehensive examination of students responses to the senior peer-facilitated approach has been provided elsewhere [37]. It reveals the vast majority (86.8%) of participants agreed or strongly agreed that the senior student facilitator made it easier for them to perform the tests. When asked to rank who they would like to administer fitness tests from most preferred to least preferred, 52.8% of students indicated that they would be in favour of the peer-assessed format used in the Youth-fit test battery, in comparison to an external expert (27.0%) or their teacher (20.2%) recording test scores [37].

Test administrators conducted several familiarisation trials, and examples of correct and incorrect trials were demonstrated. Test items included; body mass index (BMI); 20 m shuttle run test (20 m SRT); handgrip strength; standing broad jump (SBJ); 4 x 10 m shuttle run. The scientific rationale for the selection of the tests was based on their feasibility and reliability for administration in a school setting [38], and their established criterion-related validity [39]. Four additional tests of physical fitness and health commonly administered in school-based HRPF test batteries and large-scale health surveys, were also included, namely: 90° push-up; isometric plank-hold; back-saver sit-and-reach (BSR); and blood pressure. O'Keeffe and colleagues [40] confirmed the test-retest reliability of the administration protocol outlined above for each test item, reporting intra-class correlation coefficients of $\geq .797$ and mean coefficient of variation values of 6.5% across all test items. Detailed test administration protocols for each test item are available in this study [40].

All tests were conducted in participating schools' physical education halls, and took place during timetabled physical education. The authors were keen for testing to reflect the authenticity of a standard double class period of physical education in the Republic of Ireland school setting, therefore, testing lasted 80 minutes. Tests were administered in small groups of six or less participants at a testing station at any one time. Furthermore, in an effort to address fatigue or test sequencing as potential sources of measurement error, all participants had a minimum rest period of between three and five minutes between each testing station. The 20 m SRT is an estimate of maximal aerobic capacity, therefore, it was conducted on a separate day to all other tests using the Léger et al. protocol [41]. Participants were required to run between two lines 20 metres apart, while keeping pace with audio signals emitted from a pre-recorded CD. The initial speed was 8.5 km/h, and was increased by 0.5 km/h per minute. The test finished when the participant failed to reach the end lines concurrent with the audio signals on two consecutive occasions, or when the subject stopped because of fatigue.

## Data collection and quality control

A software platform was developed specifically for the purpose of this study to enable efficient multi-site capture of data from participating schools. Following test administration, cooperating physical education teachers uploaded test results to a web-based application hosted on a secured server at the lead authors' institution. Cooperating teachers received a tutorial on using the software from the lead author and a user manual outlining the procedure for inputting results. Biologically plausible value limits were assigned to each field to minimise potential

inaccuracies during data input. An additional quality control feature included collecting test battery results sheets from participating schools, from which the lead author randomly selected half of the completed results sheets, and cross-referenced each to ensure the accuracy of results inputted.

## Statistical analysis

Complete cases ($n$ = 1215; designated disadvantaged, n = 221) were extracted from the software platform and transferred to Statistical Package for Social Sciences (SPSS version 25, Chicago IL) for analysis. The research team defined an incomplete case as missing the BMI recording, or two or more fitness test items. Incomplete responses (n = 66) were excluded from all analyses. A visual inspection of histograms and box plots showed that data were normally distributed, with skewness of $\leq$ 1.2 and kurtosis of $\leq$ 1.6. Means (M) and standard deviations (SD) were calculated for all scale scores, with $t$-tests and ANOVAs testing for differences by selected demographics. A linear mixed model analysis of variance was conducted to examine the differences between designated disadvantaged and non-disadvantaged schools for key health-related fitness outcome variables, controlling for age and gender as fixed effects, and school as a random effect. Results from this analysis were then graphically depicted using clustered error bar graphs.

Following a request, access was provided to the original dataset from the Healthy Lifestyle in Europe by Nutrition in Adolescence (HELENA) study [33]. This dataset contained gender and age-specific physical fitness normative values among European adolescents from 10 European countries with a similar gender (n = 911, girls = 544) and age (13.5 years, SD .31) profile to participants in the current study. Data collection for the HELENA study took place from 2006 to 2008. Administration protocols for the five field-based tests compared, namely, the 20 m SRT, SBJ, handgrip strength, BSR, and 4 x 10 m shuttle run, were the same for both studies, as detailed in Ortega et al. [33]. Cohen's d was used to compare differences between participants in this study and age-matched European normative values by calculating the mean difference between the two groups, and dividing the result by the pooled standard deviation [33]. Cohen suggested that d = 0.2 be considered a 'small' effect size, 0.5 represents a 'medium' effect size and 0.8 a 'large' effect size. Correction for multiple comparisons was made via the Bonferroni correction. Participants physical fitness scores were also expressed using a quintile classification framework based on European normative values [33], corresponding to "very low", "low", "moderate", "high", and "very high" levels as recommended by Tomkinson et al. [42].

## Results

Anthropometric characteristics and HRPF levels of the study sample are shown in Table 2. Overall, boys had significantly higher cardiorespiratory endurance (CRE) (20 m shuttle run test), muscular fitness (handgrip strength, standing broad jump (SBJ), 90˚ push-up and isometric plank hold) levels compared to girls, while girls had significantly higher flexibility (back-saver sit-and-reach (BSR)) ($p < 0.001$, $t$-test, with Bonferroni correction for multiple comparisons). Girls had significantly lower mean systolic blood pressure in comparison to boys, however, despite reaching statistical significance, the total difference in mean values was small ($< 1.5$ mmHg). The prevalence of overweight and obesity was estimated as per the criteria published by Cole et al. [43]. Over one quarter (25.8%) of girls and 23.9% of boys were overweight, of which 12.2% of girls and 9.2% of boys were obese. An inverse relationship between performance in the 20 m SRT (r = -.32, $p < 0.001$) and SBJ (r = -.29, $p < 0.001$) tests and overweight and/or obesity was observed. Boys and girls categorised as overweight or obese

**Table 2. Descriptive characteristics of the study sample, by gender.**

| Variable | Total | Total mean (SD) | Boys | Boys mean (SD) | Girls | Girls mean (SD) | *p* value |
|---|---|---|---|---|---|---|---|
| **Age** | 1215 | 13.4 (0.4) | 606 | 13.5 (0.4) | 609 | 13.4 (0.4) | NS |
| **BMI** | 1215 | 20.3 (3.6) | 606 | 20.1 (3.6) | 609 | 20.4 (3.7) | NS |
| **BSR (cm)** [a] | 1177 | 23.5 (9.3) | 591 | 22.0 (8.7) | 586 | 25.0 (9.7) | < 0.001 [c] |
| **Systolic BP (mmHg)** | 1189 | 109.4 (13.3) | 595 | 110.7 (13.3) | 594 | 108.2 (13.1) | < 0.001 [c] |
| **Diastolic BP(mmHg)** | 1189 | 74.0 (11.4) | 595 | 73.0 (11.3) | 594 | 75.0 (11.5) | < 0.001 [b] |
| **Standing broad jump (cm)** | 1206 | 151.0 (26.1) | 601 | 158.3 (27.3) | 605 | 146.6 (23.7) | < 0.001 [b] |
| **Handgrip strength (kg)** [a] | 1201 | 23.0 (5.1) | 598 | 24.1 (5.7) | 603 | 21.9 (4.3) | < 0.001 [b] |
| **90˚ push-up (repetitions)** | 1177 | 10.9 (8.6) | 583 | 13.3 (8.6) | 594 | 8.5 (8.0) | < 0.001 [b] |
| **Isometric plank-hold (s)** | 1177 | 77.7 (49.5) | 581 | 86.7 (54) | 596 | 68.9 (42.8) | < 0.001 [b] |
| **4 x 10 m shuttle run (s)** | 1174 | 12.2 (1.4) | 588 | 12.0 (1.1) | 586 | 12.4 (1.7) | < 0.001 [b] |
| **20 m SRT (# shuttles)** | 1138 | 47.4 (22.4) | 570 | 53.4 (22.8) | 568 | 41.4 (20.3) | < 0.001 [b] |

Abbreviations: BMI, body mass index; BSR, back-saver sit and reach; SD, standard deviation; BP, blood pressure; 20 m SRT, 20 m shuttle run test.

[a] The average of right and left side scores is shown in the table. Significant differences (*p* < 0.001, with Bonferroni correction) were found between boys and girls, independent samples *t*-test.

[b] Indicates a more favourable HRPF score for boys.

[c] Indicates a more favourable HRPF score for girls. Due to school absences and/or injury, not all totals amount to 609 (girls) and 606 (boys).

ran an average of 17 fewer shuttles (20 m SRT) and jumped (SBJ), on average, 14 centimetres (cm) less than their peers.

No statistically significant differences between urban and rural schools were found across any of the variables analysed. However, differences were observed between participants in mixed-gender and single-gender schools. An ANOVA was conducted to examine differences in HRPF between boys and girls in single-gender and mixed-gender schools, using a Bonferroni adjusted *p* value of .01 for the variables listed in Table 3. Boys in mixed-gender schools had significantly higher BSR, SBJ and 20 m SRT, and significantly lower BMI levels, in comparison to participants in boys' schools. Girls in mixed-gender schools had significantly higher SBJ and handgrip strength scores in comparison to participants in girls' schools.

**Table 3. A comparison of selected health-related physical fitness variables among participants in single-gender and mixed-gender schools.**

| Variable | Mixed-gender, boys (n = 443) | Single-gender, boys (n = 163) | *p* value | Effect size $n^2$ | Mixed-gender, girls (n = 413) | Single-gender, girls (n = 196) | *p* value | Effect size $n^2$ |
|---|---|---|---|---|---|---|---|---|
| **BMI** | 19.9 (3.4) | 20.9 (4.0) | 0.002 | .02 | 20.3 (3.6) | 20.6 (3.9) | NS | .00 |
| **BSR (cm)** [a] | 23.9 (8.0) | 21.3 (8.8) | 0.002 | .02 | 26.7 (8.9) | 21.5 (10.3) | < 0.001 | .06 |
| **Standing broad jump (cm)** | 161.7 (27.2) | 150.7 (25.7) | < 0.001 | .03 | 148.6 (23.8) | 142.1 (22.1) | 0.001 | .02 |
| **Handgrip strength (kg)** [a] | 24.1 (6.2) | 24.3 (5.7) | NS | .00 | 22.3 (4.3) | 21.2 (4.1) | < 0.001[b] | .02 |
| **4 x 10 m shuttle run (s)** | 11.9 (1.0) | 12.0 (1.2) | NS | .00 | 12.4 (1.2) | 12.2 (2.3) | NS | .00 |
| **20 m SRT (# shuttles)** | 55.4 (21.7) | 48.0 (24.9) | < 0.001 | .02 | 42.2 (18.4) | 39.8 (23.6) | NS | .00 |

Data are shown as means with standard deviation in brackets. Abbreviations: BMI, body mass index; BSR, back-saver sit and reach; 20 m SRT, 20 m shuttle run test.

[a] The average of right and left side scores is shown in the table. Significant differences (*p* < 0.001, with Bonferroni correction) were found between boys and girls, independent samples *t*-test.

[b] Indicates a more favourable HRPF score for boys.

[c] Indicates a more favourable HRPF score for girls. Due to school absences and/or injury, not all totals amount to 609 (girls) and 606 (boys).

**Table 4. Descriptive characteristics of students in non-disadvantaged and designated disadvantaged schools, by gender.**

| | Boys non-disadvantaged (n = 472) | Boys disadvantaged (n = 134) | Girls non-disadvantaged (n = 522) | Girls disadvantaged (n = 87) |
|---|---|---|---|---|
| Variable | Mean (SD) | Mean (SD) | Mean (SD) | Mean (SD) |
| BMI | 19.8 (3.4) | 21.3 (4.0) | 20.2 (3.5) | 21.8 (4.2) |
| BSR (cm) [a] | 22.1 (8.5) | 21.6 (9.4) | 25.5 (9.4) | 21.4 (10.6) |
| Systolic BP (mmHg) | 110.8 (13.4) | 110.1 (13.1) | 108.2 (13.1) | 108.4 (13.5) |
| Diastolic BP (mmHg) | 72.8 (11.3) | 73.6 (11.5) | 75.1 (11.6) | 74.3 (10.6) |
| Standing broad jump (cm) | 161.8 (26.7) | 148.0 (26.5) | 147.6 (23.3) | 139.9 (23.5) |
| Handgrip (kg) [a] | 24.6 (5.5) | 22.6 (5.9) | 22 (4.1) | 21.9 (5) |
| 90˚ push-up (reps) | 14.2 (8.4) | 10.2 (8.2) | 8.9 (8.2) | 6.3 (5.8) |
| Isometric plank hold (s) | 92.2 (54.6) | 67.5 (47.4) | 70.8 (43.5) | 57.5 (36.9) |
| 4 x 10 m shuttle run (s) | 11.9 (1.0) | 12.3 (1.3) | 12.3 (1.7) | 12.7 (1.3) |
| 20 m SRT (# shuttles) | 57.8 (21.3) | 38.0 (21.3) | 43.9 (20.3) | 26.3 (12.2) |

Abbreviations: BMI, body mass index; BSR, back-saver sit and reach; SD, standard deviation; BP, blood pressure; 20 m SRT, 20 m shuttle run test

[a] The average of right and left side score is shown in the table and was used for all analyses.

Descriptive characteristics of boys and girls in non-disadvantaged versus designated disadvantaged schools are presented in Table 4. A mixed model analysis of the differences between designated disadvantaged and non-disadvantaged schools for key health-related fitness outcome variables, controlling for age and gender as fixed effects, and school as a random effect, is presented in Table 5. Mean values were significantly higher in designated disadvantaged schools for BMI ($p < 0.001$, $t$-test with Bonferroni correction) and significantly lower for the 20 m SRT ($p < 0.001$), 90˚ push-up ($p = 0.014$), SBJ ($p = 0.013$) and handgrip strength ($p = 0.05$). Differences were particularly large for the 20 m SRT, with mean values for designated disadvantaged schools 19 shuttles (380 metres) fewer than non-disadvantaged schools. Clustered error mean bar graphs, with 95% confidence intervals, were used to graphically depict the differences in performance at a school-level across key HRPF fitness variables, as displayed in Fig 1.

**Table 5. Adjusted mean differences in non-disadvantaged versus designated disadvantaged schools, controlling for age and gender as fixed effects, and school as a random effect.**

| Variable | Non- Disadvantaged (n = 994) | Disadvantaged (n = 221) | Mean difference (CI)[b] | p value |
|---|---|---|---|---|
| BMI | 20.0 | 21.6 | -1.6 (-2.2 to -0.97) | < 0.001 |
| BSR (cm) [a] | 24.3 | 20.7 | 3.5 (-3.6 to 10.7) | NS |
| Systolic BP (mmHg) | 109.4 | 108.9 | 0.5 (-2.4 to 3.4) | NS |
| Diastolic BP (mmHg) | 73.9 | 73.3 | 0.6 (-2.5 to 3.7) | NS |
| Standing broad jump (cm) | 154.3 | 142.0 | 12.3 (2.9 to 21.9) | 0.013 |
| Handgrip (kg) [a] | 23.4 | 22.0 | 1.4 (0.0 to 2.7) | 0.05 |
| 90˚ push-up (reps) | 11.5 | 8.4 | 3.1 (0.7 to 5.6) | 0.014 |
| Isometric plank hold (s) | 79.4 | 60.7 | 18.7 (-0.5 to 38.0) | NS |
| 4 x 10 m Shuttle (s) | 12.1 | 12.7 | -0.6 (-1.3 to -0.0) | NS |
| 20 m SRT (# shuttles) | 51.3 | 32.2 | 19.1 (12.3 to 25.9) | < 0.001 |

Abbreviations: BMI, body mass index; BSR, back-saver sit and reach; SD, standard deviation; BP, blood pressure; 20 m SRT, 20 m shuttle run test.

[a] The average of right and left side score is shown in the table and was used for all analyses.

[b] Mean difference (95% CI) from linear mixed models, controlling for age and gender as fixed effects, and school as a random effect.

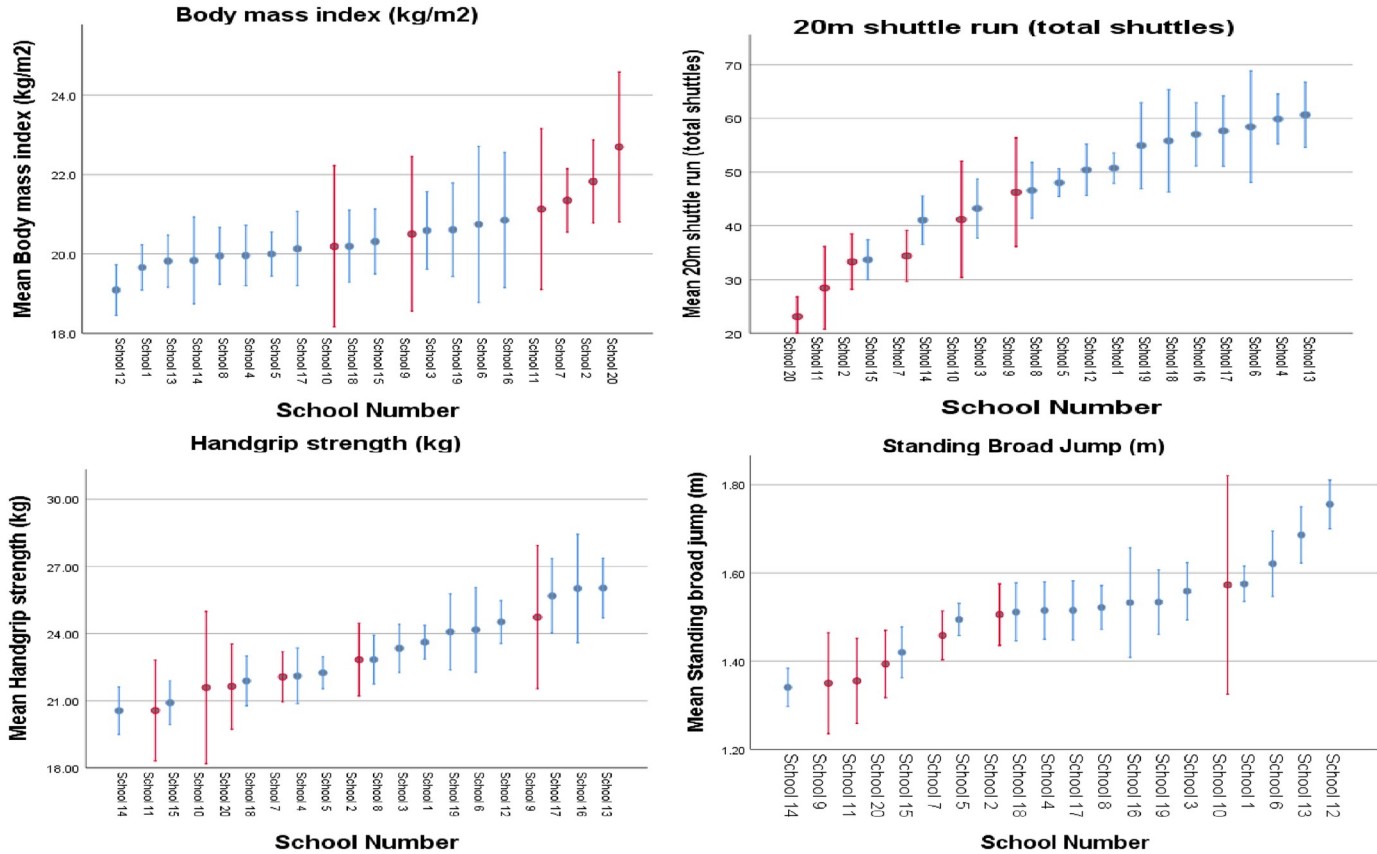

**Fig 1. Clustered error bar mean graphs of 20 m shuttle run, body mass index, standing broad jump and handgrip strength tests with 95% confidence intervals, by school.** Designated disadvantaged schools are highlighted red, non-disadvantaged schools are highlighted blue.

When compared to European normative values [33], girls in this study scored significantly better in the 20 m SRT, 4 x 10 m shuttle run and SBJ tests, while boys scored significantly higher in the BSR test (Cohen's d ranging from 0.2 to 0.6, $p < 0.01$, Bonferroni correction). However, European adolescents had significantly higher handgrip strength scores (Cohen's d 0.6 to 0.8, $p < 0.01$). European boys also had significantly higher SBJ scores (Cohen's d = 0.5, $p < 0.01$). Using a quintile classification framework, the authors established the percentage of Irish adolescents from the current study that fell within each quintile of European normative values [33] (Figs 2 and 3). Only 11.1% of boys and 10.0% of girls achieved a very high score > 80[th] centile) for handgrip strength, with 34.0% of boys and 31.2% of girls classified in the very low quintile (< 20%). SBJ scores were more evenly spread, 19.5% and 9.9% scoring in the very low category, and 17.6% and 17.1% scoring in the very high category, for boys and girls, respectively. With regard to the BSR test, 39.1% of boys and 29.2% of girls achieved a very high score based on European norms. While boys' scores for the 4 x 10 m shuttle run were relatively evenly distributed in each quintile, 41.1% of girls achieved a score ≥ 80[th] per-centile. Finally, the most significant differences with European normative data were found in the 20 m SRT. Almost two thirds of girls (61.4%) and 41.1% of boys in the current study achieved a very high ranking (≥ 80[th] percentile), with only 5.6% of boys and 0.9% of girls cate-gorised in the very low category (≤ 20[th] percentile).

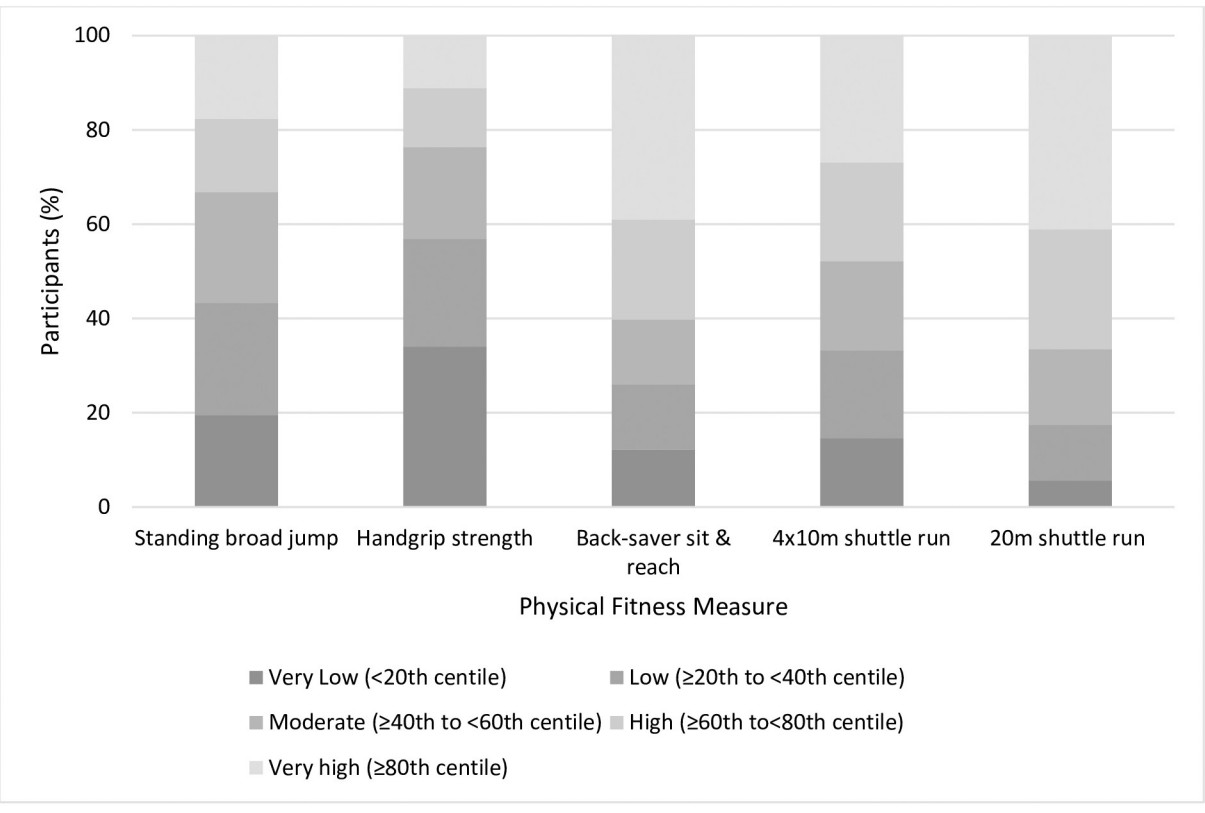

**Fig 2. Quintile classification framework for physical fitness components for boys (n = 606), based on European normative values [33].**

## Discussion

The aim of the current study was to examine the influence of school-level characteristics on fitness test performance and to compare Irish adolescents' physical fitness to European norms. Overall, participants in designated disadvantaged schools had significantly poorer HRPF levels in comparison to those in non-disadvantaged schools, participants in mixed-gender settings had significantly lower BMI levels and higher muscular strength levels compared to those in single-gender schools, and although participants in this study had significantly higher CRE levels, European adolescents had significantly higher muscular strength levels. This study represents the first analysis of all components of HRPF among adolescents from the Republic of Ireland, which we hope will form the basis of further examinations of physical fitness variables among youth across a broader range of age groups.

Perhaps unsurprisingly, boys scored higher than girls across all components of HRPF, aside from flexibility. This corroborates the findings of a recent meta-analysis of physical fitness among adolescents internationally [44], which reported that boys consistently scored higher than girls on fitness tests, except on the sit-and-reach test of flexibility in which girls scored higher. A recent national survey of physical activity and sport participation among Irish youth reported that only 7% of girls in secondary schools met the recommended daily activity guidelines of 60 minutes of moderate-to-vigorous physical activity compared to 14% of boys [30]. The authors also reported non-participation levels of 45% in any form of community sport among adolescent girls, in comparison to 31% among boys. Furthermore, in line with research to date, an inverse relationship between performance in the 20 m SRT and SBJ tests and overweight/obesity was found in the current sample. In an investigation of the determinant factors

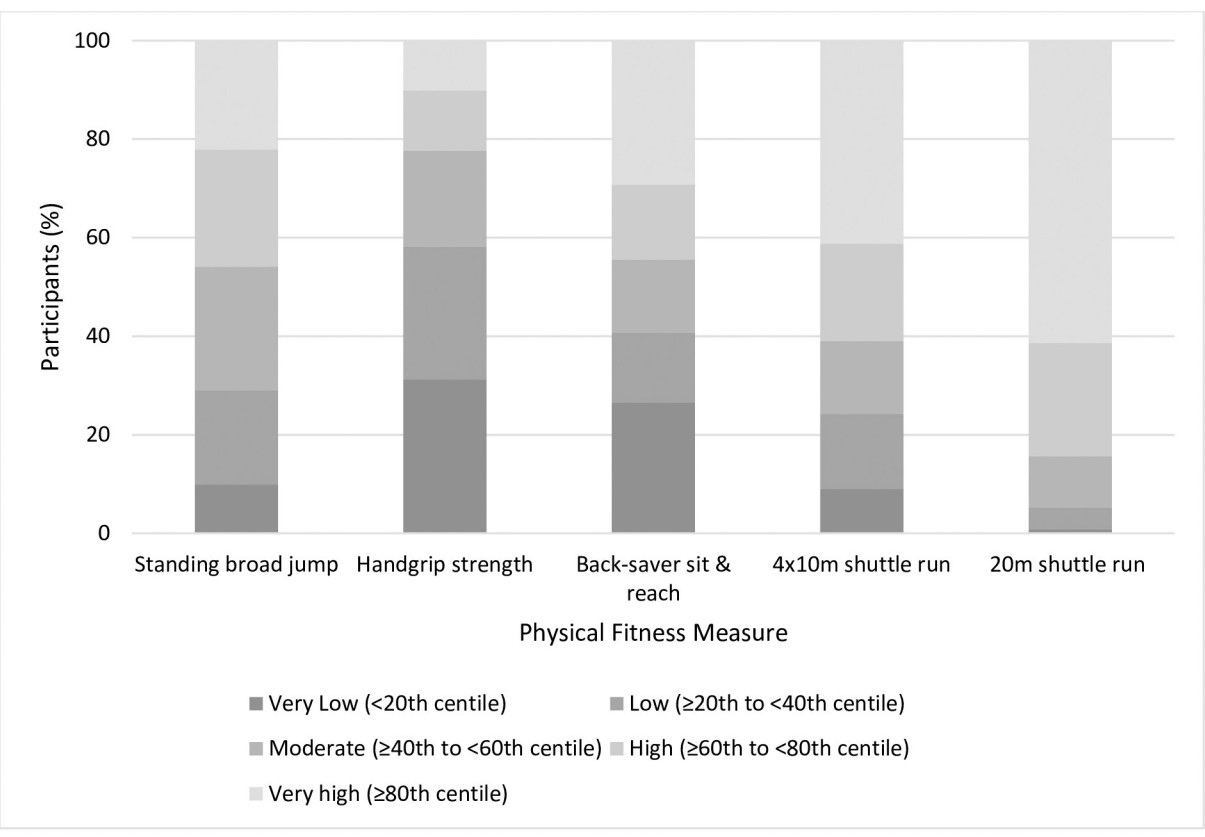

**Fig 3. Quintile classification framework for physical fitness components for girls (n = 609), based on European normative values [33].**

of physical fitness among 13,622 European children, Zaqout and colleagues [21] highlighted the significance of BMI as a physical fitness determinant, independent of physical activity. In an examination of overweight/obesity and physical fitness among 519 Brazilian children and adolescents aged 7 to 15 years, Dumith et al. [22] also reported that higher BMI values were associated with declines in physical fitness, independent of age. The prevalence of overweight, classified according to the same age and sex specific cut points [43], was 25% among a nationally representative sample of Irish adolescents in a recent longitudinal study [45], one percent less than the figure reported from data generated as part of the current study. In addition, age-matched mean 20 m SRT values were similar to those from the most recently reported national representative data [30].

Participants in mixed-gender schools had significantly higher estimated CRE (20 m SRT) and muscular strength (handgrip strength), and significantly lower BMI levels, in comparison to participants in single-gender schools. A recent survey of students attitudes towards fitness testing in school settings [37] reported that participants in mixed-gender settings had significantly more positive attitudes than those in single-gender schools. Ishee and Ward [46] did report that girls in mixed-gender schools achieved higher moderate to vigorous physical activity levels in their physical education lessons in comparison to those in single-gender schools, however, boys did not vary significantly. Van Acker et al. [47] similarly reported that participants in coeducational settings achieved significantly higher moderate to vigorous physical activity levels during physical education lessons in comparison to single-gender schools.

Contrasting findings emerged from comparisons between the current study sample and age-matched European normative values generated from the HELENA study [33]. Girls in this study had significantly higher mean scores in the 4 x 10 m shuttle run and 20 m SRT in comparison to their European peers, while boys in this study scored significantly higher in the back-saver sit-and-reach test. However, as illustrated in Figs 2 and 3, over three quarters of Irish boys and girls were classified as moderate or below average for handgrip strength when compared to European norms using a quintile classification framework. This is a finding of particular concern given the emerging evidence-base linking poor musculoskeletal fitness in adolescence with negative health outcomes later in life [4, 48]. In contrast, 84.3% of girls and 66.5% of boys scored above the 60th percentile in the 20 m SRT when compared to European normative values. Mean 20 m SRT values were similar to those from the most recent nationally representative data [30]. Nationally, research has indicated a significant drop off in sports participation and physical activity rates among young adolescent girls from the age 14 [49]. Although beyond the scope of the current study, an examination of fitness variables across all school-going adolescent age groups is needed confirm if the reported differences between Irish and European adolescents track across all adolescent age groups.

An important finding to emerge from this study was the disparity in fitness levels between participants in designated disadvantaged and non-disadvantaged schools. Participants in designated disadvantaged schools had significantly higher BMI levels, and significantly lower 20 m SRT and SBJ scores in comparison to those in non-disadvantaged schools. A comprehensive analysis on the influence of socioeconomic status on physical fitness among European adolescents concluded that socioeconomic status was positively associated with physical fitness, independently of total body fat and habitual physical activity [23]. In one of the few empirical studies to investigate the impact of school sociodemographic characteristics on physical fitness variables, Welk et al. [50] reported that physical fitness was consistently higher among students in schools categorized as low diversity and high socioeconomic status. Bai and colleagues (2016) similarly reported clear evidence that school socioeconomic status was the most influential contextual factor for explaining disparities in school fitness outcomes. It has also recently been reported that school socioeconomic status was a strong determinant of overweight and obesity in Irish schoolchildren [25]. This suggests that government funding utilised for the promotion of healthy lifestyle behaviours among youth should provide additional support for designated disadvantaged schools.

This study had some limitations which should be noted. Firstly, although the current study sample represents the largest examination of multiple components of HRPF in the Republic of Ireland to date, due to logistical constraints, participants were only generated from year one of secondary school education, precluding an analysis of fitness variables across all adolescent age groups. Additionally, the sample size of 20 schools is small for the linear mixed model analysis of non-disadvantaged (n = 14) versus designated disadvantaged (n = 6) schools. However, the randomised and stratified nature of the sample, the variety of fitness tests used and the provision of the original HELENA dataset to facilitate more detailed comparisons with participants in the current study, were important strengths.

## Conclusion

This study represents the first comprehensive review of multiple components of health-related fitness among a stratified sample of adolescents in the Republic of Ireland. The contrasting findings for different fitness components within our sample reiterate the need for multi-component HRPF test batteries for monitoring physical fitness in youth. Overall, age-matched comparisons of HRPF levels with European norms were broadly positive for all components,

aside from muscular fitness in which European adolescents scored significantly higher. Therefore, interventions aimed at improving the physical fitness and activity levels of Irish youth should include a focus on muscular fitness. In terms of school level characteristics specifically, data presented in the current study indicated that adolescents in mixed gender schools outperformed those in single gender schools across most HRPF components. Furthermore, the extent of the disparity in fitness levels between participants in designated disadvantaged and non-disadvantaged schools was a finding of particular concern. Future interventions designed to promote healthy lifestyle behaviours among school-going populations should give special consideration to students in designated disadvantaged schools. The provision of additional support funds to promote healthy lifestyle behaviours could represent an efficient model of funding, targeting those who are most in need.

## Supporting information

**S1 File.**
(SAV)

## Acknowledgments

The authors would like to acknowledge the assistance of the participants, parents, principals and, in particular, cooperating physical education teachers involved in this study. The authors would also like to acknowledge iMosphere® for their assistance in the development and design of the web-based component.

## Author Contributions

**Conceptualization:** Alan E. Donnelly.

**Formal analysis:** Ciaran MacDonncha, Helen Purtill, Alan E. Donnelly.

**Investigation:** Ciaran MacDonncha.

**Methodology:** Ciaran MacDonncha.

**Supervision:** Ciaran MacDonncha, Alan E. Donnelly.

**Writing – review & editing:** Ciaran MacDonncha, Alan E. Donnelly.

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
