## [Decision Letter · Decision Letter 0]

2 Apr 2020

PONE-D-20-05075

Profiling the health related fitness of Irish adolescents: A school level socioeconomic status divide.

PLOS ONE

Dear Mr O'Keeffe,

Thank you for submitting your manuscript to PLOS ONE. After careful consideration, we feel that it has merit but does not fully meet PLOS ONE’s publication criteria as it currently stands. Therefore, we invite you to submit a revised version of the manuscript that addresses the points raised during the review process.

Comments from two expert reviewers can be found below. Overall, the manuscript was viewed positively by both, however each have raised several important questions relating to the study rationale and comparison with other European datasets, outcome measure assessments and the analysis and reporting of the data. Please ensure all questions are fully addressed in your revision.

We would appreciate receiving your revised manuscript by May 17 2020 11:59PM. To enhance the reproducibility of your results, we recommend that if applicable you deposit your laboratory protocols in protocols.io, where a protocol can be assigned its own identifier (DOI) such that it can be cited independently in the future. For instructions see: http://journals.plos.org/plosone/s/submission-guidelines#loc-laboratory-protocols

We look forward to receiving your revised manuscript.

Kind regards,

Kathryn L. Weston, PhD

Academic Editor

PLOS ONE

2. In your Methods section, please provide additional information about the participant recruitment method and the demographic details of your participants. Please ensure you have provided sufficient details to replicate the analyses such as: a) the recruitment date range (month and year), b) a description of any inclusion/exclusion criteria that were applied to participant recruitment, c) a table of relevant demographic details, d) a statement as to whether your sample can be considered representative of a larger population, e) a description of how participants were recruited, and f) descriptions of where participants were recruited and where the research took place.

4. Please upload a new copy of Figure 1 as the detail is not clear. Please follow the link for more information: http://blogs.PLOS.org/everyone/2011/05/10/how-to-check-your-manuscript-image-quality-in-editorial-manager/

Reviewers' comments:

Reviewer's Responses to Questions

**Comments to the Author**

1. Is the manuscript technically sound, and do the data support the conclusions?

Reviewer #1: Yes

Reviewer #2: Yes

2. Has the statistical analysis been performed appropriately and rigorously? 

Reviewer #1: Yes

Reviewer #2: Yes

3. Have the authors made all data underlying the findings in their manuscript fully available?

Reviewer #1: Yes

Reviewer #2: No

4. Is the manuscript presented in an intelligible fashion and written in standard English?

Reviewer #1: Yes

Reviewer #2: Yes

5. Review Comments to the Author

Reviewer #1: An interesting paper that is clear and easy to follow. I have a few minor comments:

I think it is usual practice to report actual p values, unless p<0.001, this hasn't been done consistently.

Did you create BMIz-scores to calculate the proportions that were overweight? If so this data should go in the tables and could be used in the analysis.

Discussion, third paragraph "...physical activity rates among young ADOLESCENT girls"... not adolescents, unless you don't mean to have the word 'girls' in the sentence?

Fig 1 and 2 need to have the categories altered slightly, as at the moment the boundaries fall into two categories, ie someone on the 20th centile is currently both 'very low' and 'low'. Ditto 40th/60th/80th. Check which need to be >< or also equals.

Reviewer #2: Many thanks to the authors for their work in preparing this manuscript. I enjoyed reading the manuscript and feel it will make a good contribution to the literature, given the lack of published data for this country, employing more objective methods. I have outlined queries below.

Introduction

The authors have presented a good overview of the literature in adolescents and the different findings across these datasets. I feel this section could be further strengthened by briefly offering explanations for these differences – are these attributed to methodological differences in how outcomes were measured, inconsistencies in reference points applied, or as a result of the samples recruited.

Methods

The authors could provide some further justification to compare with the European dataset. The methodology should make reference to the population in the European dataset (similar age, gender) and if outcomes were measured in the same way or how any differences were accounted for in the analysis.

Data collection and quality control – I was interested if the schools noted or reported any issues with participants being measured by older peers? I feel the authors should consider the rationale for this approach, and whether it may have introduced a bias within the study sample. Is it possible that some participants (perhaps those who were less sporty/fit/overweight) may have been put off from taking part because of this data collection approach, and how this may have impacted upon the results across schools?

The authors note that disadvantaged schools make up ~ 25% of all secondary schools. Given the difference in sample size across non-disadvantaged and disadvantaged within the results, did the authors consider recruiting a higher proportion of disadvantaged schools to provide a more balanced sample across groups.

Results

Lines 231 – 234 – These findings are quite descriptive and the authors should avoid highlighting individual mean scores.

From Table 2, were any other school level factors (mixed-gender / school location) considered in the analysis which may have influenced differences between disadvantaged and non- disadvantaged schools.

Discussion (line numbers missing)

The opening paragraph highlights that the aim was to look at school level characteristics, and first highlighted results discusses males vs females. Was this technically a school level characteristic i.e. did this only compare all male vs all female schools? I feel this could be further clarified, as it is more a characteristic of the overall gender breakdown sample as opposed to a school level characteristic?

Paragraph 2 notes the higher scores for boys across the sample – did the authors consider collecting any additional information on physical activity levels/sports participation. Would suggest including some references here to support the reasoning behind these observed findings.

Minor comments

Manuscript uses males and females in some instances, then boys and girls in others. Would suggest a consistent approach across the manuscript.

Abstract

Background: Include introductory line to provide context to the article before outlining the study aims

Method: Line 21 – Reword to clarify that tests were undertaken to measure components of physical fitness

Main body

Methods

Were all tests performed in the same order by each study participant to control for the potential of some testing to impact upon performance in another outcome?

Results

Of the 20 schools recruited, how many were disadvantaged? Can only see student numbers, not school numbers also.

What was the mean number of students who participated across each school?

Formatting

Appears to be inconsistency in font size across manuscript version

The figures are unclear/blurred and difficult to view

6. PLOS authors have the option to publish the peer review history of their article (what does this mean?). If published, this will include your full peer review and any attached files.

Reviewer #1: No

Reviewer #2: No

---

## [Author Response · Author response to Decision Letter 0]

20 Apr 2020

Response to Reviewers 

(PONE-D-20-05075)

The authors would like to thank the editor and reviewers for their time in reviewing this manuscript. Each comment is addressed in the comment log below, and highlighted in red font colour on the updated manuscript.

Response to editor comments

Response: Formatting updates have been made to ensure the latest submission aligns with style requirements. 

2. In your Methods section, please provide additional information about the participant recruitment method and the demographic details of your participants. Please ensure you have provided sufficient details to replicate the analyses such as: 

a) The recruitment date range (month and year),

Response: Manuscript updated P.7 L.157-159. 

b) A description of any inclusion/exclusion criteria that were applied to participant recruitment,

Response: Manuscript updated. Schools: P.6 L.119-121 and P.7 L.141 -143. Student participants: P.7 L3.147-152. 

c) A table of relevant demographic details,

Response: The manuscript has been updated to include a demographic profile of participants, see Table 1 P.8 L.160-164. 

d) A statement as to whether your sample can be considered representative of a larger population,

Response: Manuscript updated: P.6 L136-139. 

e) A description of how participants were recruited,

Response: The is detailed on P.7 L.144-147. 

f) Descriptions of where participants were recruited and where the research took place.

Response: The manuscript has been updated to provide clarification. Where participants were recruited: P.7 L.140-159. Where the research took place: P.9 L.192-198.

3. We note that you have indicated that data from this study are available upon request. PLOS only allows data to be available upon request if there are legal or ethical restrictions on sharing data publicly.

Response: Following consultation and approval from our Institution’s Research Ethics Review Board, the de-identified dataset will now be made available. 

4. Please upload a new copy of Figure 1 as the detail is not clear. Please follow the link for more information: http://blogs.PLOS.org/everyone/2011/05/10/how-to-check-your-manuscript-image-quality-in-editorial-manager/

Response: Figure 1 has been updated using PACE software as recommended. 

5. While revising your submission, please upload your figure files to the Pre-flight Analysis and Conversion Engine (PACE) digital diagnostic tool, https://pacev2.apexcovantage.com/. PACE helps ensure that figures meet PLOS requirements. To use PACE, you must first register as a user. Registration is free. Then, login and navigate to the UPLOAD tab, where you will find detailed instructions on how to use the tool. If you encounter any issues or have any questions when using PACE, please email us at figures@plos.org. Please note that Supporting Information files do not need this step.

Response: All three figures have been updated using PACE software, and are attached on the revised submission. 

Response to reviewer one comments

1. An interesting paper that is clear and easy to follow. I have a few minor comments:

Response: The authors would like to thank you for your time in conducting this review. Each comment is addressed in the comment log below, and all edits are highlighted in red font colour on the updated manuscript. 

2. I think it is usual practice to report actual p values, unless p<0.001, this hasn't been done consistently. 

Response: Thank you for your recommendation. Where appropriate, p values have been updated to actual values throughout the manuscript, unless p <0.001 as you indicate above. Occasionally, where multiple individual variables produced p values of < 0.001, they are referred to as a collective rather than listing all individual p values. e.g. Abstract P.1 L.32-33. 

3. Did you create BMI z-scores to calculate the proportions that were overweight? If so this data should go in the tables and could be used in the analysis. 

Response: Thank you for your comment. BMI z-scores were not calculated. The prevalence of overweight and obesity was estimated as per the criteria published by Cole et al. (2000) as indicated on P.12 L.254-255. These set threshold values were used by the most recent national survey of health among a national sample, which enabled a direct comparison. See P.18 L.364 -369. 

4. Discussion, third paragraph "...physical activity rates among young ADOLESCENT girls"... not adolescents, unless you don't mean to have the word 'girls' in the sentence?

Response: Updated to “adolescent girls”. 

5. Fig 1 and 2 need to have the categories altered slightly, as at the moment the boundaries fall into two categories, i.e. someone on the 20th centile is currently both 'very low' and 'low'. Ditto 40th/60th/80th. Check which need to be >< or also equals.

Response: Thank you for recognising this mistake. This was an oversight on the authors’ behalf. Figures 2 and 3 have been updated to include the correct symbols. 

Response to reviewer two comments

1. Many thanks to the authors for their work in preparing this manuscript. I enjoyed reading the manuscript and feel it will make a good contribution to the literature, given the lack of published data for this country, employing more objective methods. I have outlined queries below.

Response: The authors would like to thank your time in providing such detailed recommendations and sagacious advice on the manuscript. Each comment is addressed in the comment log below, and all edits are highlighted in red font colour on the updated manuscript. 

2. Introduction

The authors have presented a good overview of the literature in adolescents and the different findings across these datasets. I feel this section could be further strengthened by briefly offering explanations for these differences – are these attributed to methodological differences in how outcomes were measured, inconsistencies in reference points applied, or as a result of the samples recruited.

Response: Thank you for your recommendation. The authors agree that this is an important point to address in light of the stated aims of the manuscript. Therefore, the introduction has been updated to include a new paragraph that details the importance of interpreting and comparing datasets with caution. See P.4 L.66-83. 

3. Methods

The authors could provide some further justification to compare with the European dataset. The methodology should make reference to the population in the European dataset (similar age, gender) and if outcomes were measured in the same way or how any differences were accounted for in the analysis.

Response: Thank you for your suggestion. The statistical analysis paragraph of the methodology section has been updated to provide a more comprehensive overview of the demographic profile and protocols used in the HELENA study, thus forming a stronger rationale to conduct the comparison. See P.11 L.230-237. 

4. Data collection and quality control 

I was interested if the schools noted or reported any issues with participants being measured by older peers? I feel the authors should consider the rationale for this approach, and whether it may have introduced a bias within the study sample. Is it possible that some participants (perhaps those who were less sporty/fit/overweight) may have been put off from taking part because of this data collection approach, and how this may have impacted upon the results across schools?

Response: Thank you for your query. A comprehensive examination of students responses to the senior peer-facilitated methodological designed utilised in the current manuscript is provided in a separate peer reviewed article in the European Physical Education Review Journal. This study reveals that the vast majority (86.8%) of students agreed or strongly agreed that the senior student facilitator made it easier for them to perform the tests. When asked to rank who they would like to administer fitness tests from most preferred to least preferred, 52.8% of students indicated that they would be in favour of the senior-peer facilitated format used in the Youth-fit test battery, in comparison to an external expert (27.0%) or their teacher (20.2%) recording test scores. P.8 L.173-179 has been updated to reflect this. 

In terms of the potential impact of this approach on participation rates, participation rates in the final sample were ≥75% in all schools, with a mean of 86% per school, and many schools reached maximum participation. It should also be stated, as noted on P.8 L.171-173 that senior peer-facilitators participated in a three hour workshop on test administration protocols. The importance of administering the tests sensitively and in a supportive manner were addressed during this workshop. Finally, as mentioned on P.18 L.364-368, the prevalence of overweight and estimated cardio-respiratory fitness were very similar to most recently reported nationally representative data, further emphasising the potential for this approach to monitoring key health indicators on a larger-scale. 

5. The authors note that disadvantaged schools make up ~ 25% of all secondary schools. Given the difference in sample size across non-disadvantaged and disadvantaged within the results, did the authors consider recruiting a higher proportion of disadvantaged schools to provide a more balanced sample across groups. 

Response: The objective was to recruit a sample from the mid-west and south-west region of Ireland that was representative of national demographic characteristics, hence, 6 designated disadvantaged schools were recruited representing 30% of schools in the final sample. It should also be noted that, on average, student numbers in designated disadvantaged schools are significantly lower than non-designated schools. 

6. Results

Lines 231 – 234 – These findings are quite descriptive and the authors should avoid highlighting individual mean scores.

Response: Thank you for your recommendation. The authors agree that this is overly descriptive and does not add to the paper. We have removed these lines in the updated manuscript. 

7. From Table 2, were any other school level factors (mixed-gender / school location) considered in the analysis which may have influenced differences between disadvantaged and non- disadvantaged schools.

Response: The results section has been updated to indicate that no statistically significant differences were found between urban and rural schools for any of the variables analysed. P.12 L.261-262. 

8. Discussion (line numbers missing)

Response: Apologies for this oversight. The updated manuscript includes continuous line numbers for the entire manuscript. 

9. Discussion: The opening paragraph highlights that the aim was to look at school level characteristics, and first highlighted results discusses males vs females. Was this technically a school level characteristic i.e. did this only compare all male vs all female schools? I feel this could be further clarified, as it is more a characteristic of the overall gender breakdown sample as opposed to a school level characteristic?

Response: Thank you for your comment. The authors acknowledge that the gender comparison does not specifically address the stated aim (i.e. school characteristics), and thus, have removed reference to this comparison from the opening paragraph of the discussion. The authors do feel that there is merit in the second paragraph of the discussion which provides an overview of male and female participants given the dearth of HRPF data specific to this population. The authors have also updated the results section to provide a comparison between males and females in single-gender and mixed-gender schools. This is a school level characteristic which we feel is of direct relevance to the stated aim, and would be of specific interest to the target readership. See P.12 L.262-268. Also, see Table 3 P.13 L.276-282. The discussion section has also been updated to reference the school gender comparison (P.20 L.365-376). 

10. Paragraph 2 notes the higher scores for boys across the sample – did the authors consider collecting any additional information on physical activity levels/sports participation. Would suggest including some references here to support the reasoning behind these observed findings.

Response: Thank you for your comment. The collection of additional information beyond HRPF was not an objective of the current study. However, the authors acknowledge that the inclusion of such data could serve to explain some of the gender disparities. Therefore, the manuscript has been updated to include data from a recent national survey of physical activity and sport participation levels among youth. (P.18 L.369-380)

Minor comments

11. General

Manuscript uses males and females in some instances, then boys and girls in others. Would suggest a consistent approach across the manuscript.

Response: The manuscript has been updated to ensure consistency throughout when referencing participant gender. 

12. Abstract

Background: Include introductory line to provide context to the article before outlining the study aims

Response: Thank you for this suggestion. The introductory line of the abstract has been updated to provide context of the article (P.1 L15-16) “Examining factors that may explain disparities in fitness levels among youth is a critical step in youth fitness promotion.”

13. Method: Line 21 – Reword to clarify that tests were undertaken to measure components of physical fitness

Response: The methods section of the abstract has been updated to clarify that tests were undertaken to measure the components of health related physical fitness. (P.1 L.22) 

14. Main body

Methods

Were all tests performed in the same order by each study participant to control for the potential of some testing to impact upon performance in another outcome?

Response: P.9 L.192-198 has been updated to reflect test sequencing. Tests were not performed in the same order by all participants due to time constraints. The authors were keen for testing to reflect the authenticity of a standard school environment, therefore, testing lasted 80 minutes, which is a standard double class period of physical education in the Republic of Ireland. Tests were administered in small groups with no more than six students at a testing station at any one time. Furthermore, in an effort to address potential participant fatigue and/or test sequencing as potential sources of error, the test battery was sequenced to ensure participants had a minimum rest period of three to five minutes between each testing station. 

15. Results

Of the 20 schools recruited, how many were disadvantaged? Can only see student numbers, not school numbers also.

Response: Table 1 in the updated manuscript provides a full demographic profile of the sample, including the number of schools for each of the chosen strata. 

16. What was the mean number of students who participated across each school?

Response: The manuscript has been updated to include the mean number of participants across each school (P.7 L.157-159)

Formatting

17. Appears to be inconsistency in font size across manuscript version

Response: The manuscript has been updated to ensure consistency in font size throughout. 

18. The figures are unclear/blurred and difficult to view

Response: In line with the recommendations from the editor, all three figures in the latest manuscript have been updated using PLOS One’s online image quality enhancer tool PACE.

---

## [Decision Letter · Decision Letter 1]

12 Jun 2020

Profiling the health-related physical fitness of Irish adolescents: A school-level sociodemographic divide.

PONE-D-20-05075R1

Dear Dr. O'Keeffe,

We’re pleased to inform you that your manuscript has been judged scientifically suitable for publication and will be formally accepted for publication once it meets all outstanding technical requirements.

Kind regards,

Kathryn L. Weston, PhD

Academic Editor

PLOS ONE

Additional Editor Comments (optional):

Please note some very small comments from the Reviewers regarding a minor typo in Table 3, spacing throughout the manuscript and the suggestion to remove a sentence from the Discussion. Please ensure these are addressed in the final version of the manuscript. 

Reviewers' comments:

Reviewer's Responses to Questions

**Comments to the Author**

1. If the authors have adequately addressed your comments raised in a previous round of review and you feel that this manuscript is now acceptable for publication, you may indicate that here to bypass the “Comments to the Author” section, enter your conflict of interest statement in the “Confidential to Editor” section, and submit your "Accept" recommendation.

Reviewer #1: All comments have been addressed

Reviewer #3: (No Response)

2. Is the manuscript technically sound, and do the data support the conclusions?

Reviewer #1: (No Response)

Reviewer #3: Yes

3. Has the statistical analysis been performed appropriately and rigorously? 

Reviewer #1: (No Response)

Reviewer #3: Yes

4. Have the authors made all data underlying the findings in their manuscript fully available?

Reviewer #1: (No Response)

Reviewer #3: No

5. Is the manuscript presented in an intelligible fashion and written in standard English?

Reviewer #1: No

Reviewer #3: Yes

6. Review Comments to the Author

Reviewer #1: The additions have really strengthened the paper.

Check the new table 3 as one of the variables has two decimal places (this is the minor revision, as PLOS ONE says it does not correct typos).

Reviewer #3: Thank you for addressing the comments of the reviewer. I have a few minor comments/amendments required. Please check the full manuscript for adherence and consistency to the SI units of reporting. For example, for 20m shuttle run, there should be a space between the number and unit. Please check all numbers and units. Additionally the authors sometimes use N in capitals, other times as n. Please ensure there is a space between n and the equals sign. For example, n = 35. Not n= 35. Finally, within the discussion, the authors state....to the best of the authors knowledge this is the first study.......I would personally consider refraining from using such expressions. Firstly, you cannot be sure. And secondly, I am not sure what this adds to the manuscript and the interpretation of the findings. It may perhaps be better to allow the reader to decide. A short article on why to avoid such phrases can be found here https://www.thelancet.com/journals/lancet/article/PIIS0140-6736(04)17096-7/fulltext

7. PLOS authors have the option to publish the peer review history of their article (what does this mean?). If published, this will include your full peer review and any attached files.

Reviewer #1: No

Reviewer #3: No

---

## [Editor Report · Acceptance letter]

19 Jun 2020

PONE-D-20-05075R1 

Profiling the health-related physical fitness of Irish adolescents: A school-level sociodemographic divide. 

Dear Dr. O'Keeffe:

I'm pleased to inform you that your manuscript has been deemed suitable for publication in PLOS ONE. Congratulations! Your manuscript is now with our production department. 

Kind regards, 

on behalf of

Dr. Kathryn L. Weston 

Academic Editor

PLOS ONE